# Recent Advances in *N*-Heterocyclic Small Molecules for Synthesis and Application in Direct Fluorescence Cell Imaging

**DOI:** 10.3390/molecules28020733

**Published:** 2023-01-11

**Authors:** Yanan Li, Tao Liu, Jianan Sun

**Affiliations:** 1School of Biomedical Engineering, Anhui Medical University, Hefei 230032, China; 2School of Chemical Engineering, Anhui University of Science and Technology, Huainan 232001, China

**Keywords:** nitrogen-containing heterocycles, organic small molecules, synthesis, fluorescence cell imaging

## Abstract

Nitrogen-containing heterocycles are ubiquitous in natural products and drugs. Various organic small molecules with nitrogen-containing heterocycles, such as nitrogen-containing boron compounds, cyanine, pyridine derivatives, indole derivatives, quinoline derivatives, maleimide derivatives, etc., have unique biological features, which could be applied in various biological fields, including biological imaging. Fluorescence cell imaging is a significant and effective imaging modality in biological imaging. This review focuses on the synthesis and applications in direct fluorescence cell imaging of *N*-heterocyclic organic small molecules in the last five years, to provide useful information and enlightenment for researchers in this field.

## 1. Introduction

Nitrogen-containing heterocycles are cyclic molecules with one or more nitrogen atoms in the cyclic scaffold, which are important and unique classes among heterocycles [1]. Nitrogen-based heterocyclic structures are ubiquitous in biologically active natural products, pharmaceutical drugs, and agrochemicals [2,3,4]. According to the statistics, more than 59% of the FDA-approved small-molecule drugs bear at least one nitrogen heterocycle [5]. Moreover, nitrogen-containing heterocycles possess various physiological and pharmacological properties [6,7].

Magnetic resonance imaging (MRI), positron emission tomography (PET), ultrasound imaging (US), and radionuclides have been studied and applied for biological diagnoses [8]. These imaging techniques have great significance in the bioimaging field; however, they still have some shortcomings, such as: poor accuracy, low spatiotemporal resolution, and high radiation risk [9,10]. Fluorescence imaging (FI), as an attractive imaging technique, has many advantages, including high selectivity, high sensitivity, low cost, in situ real-time detections, non-invasive, and non-radiative characteristics [11,12]. It is a simple but effective imaging approach that can be applied to the in-depth study of the physiological and pathological processes at molecular, cellular, and tissue levels in multiple dimensions and in real time [13]. Several *N*-heterocyclic fluorophores such as indocyanine green and methylene blue have been approved by the FDA for clinical use and have been utilized for fluorescence-guided tumor resection during clinical surgery [14].

Organic small molecular dyes have many advantages, such as easy modification and functionalization, easy metabolism, low toxicity, good biocompatibility, and tailored optical properties [15]. The previous reviews mainly focused on the drug treatment of *N*-heterocyclic organic small molecules, and the reviews on imaging are few. Therefore, the fluorescence characteristics of *N*-heterocyclic organic small molecules have received our interest.

In this review, we mainly introduce recent advances since 2018 in direct fluorescence cell imaging of nitrogen-containing heterocyclic organic small molecules, including nitrogen-containing boron compounds, cyanine, pyridine derivatives, indole derivatives, quinoline derivatives, maleimide derivatives, and others (Figure 1). This review is divided into seven sections according to the different scaffolds of nitrogen-containing heterocycles, selected representative examples are described schematically, and finally, the challenge of direct fluorescence cell imaging of *N*-heterocyclic small molecules is discussed.

## 2. Nitrogen-Containing Boron Compounds

### 2.1. Boron Dipyrromethene (BODIPY) Derivates

The 4,4′-difluoro-4-bora-3*a*,4*a*-diaza-s-indacene dyes, usually abbreviated as BODIPYs, are one of the most significant fluorescent dyes [16]. The first BODIPY dye was synthesized by chemists Treibs and Kreuzer in 1968 [17]. Since BODIPYs offer unique properties, including high photostability, high fluorescence quantum yields, easy modification, and other advantages, the BODIPY derivates have attracted the attention of chemical and biological researchers in recent years [18]. Here, several representative BODIPY derivates applied for direct live-cell fluorescence imaging were introduced.

In 2019, Hao’s group developed a novel BODIPY with near-infrared (NIR) absorption and bright fluorescence [19]. Various BODIPYs were synthesized through a metal- and additive-free direct C−H α-arylation of easily accessible BODIPYs with aryl diazonium salts in the presence of visible light (Figure 1). Representative diindole-annulated BODIPY **1** shows a maximum absorption peak and a maximum emission peak at the NIR region, respectively. The CCK-8 assay indicated that BODIPY **1** has good biocompatibility with cells. Fluorescence images stained with BODIPY **1** revealed bright red fluorescence in the cytoplasm of HeLa cells (Figure 2). The BODIPY core was expanded with nearly planar annulate indoles, thus resulting in a well-extended π-conjugation and a red shift of the absorption and emission.

In 2020, Tang et al. synthesized a new quinoline-fused BODIPY **2** [20]. This quinoline-fused BODIPY **2** was obtained in moderate yield by oxidation of precursory BODIPY with FeCl_3_ (Figure 2). Due to the fusion of the quinoline group, the expansion of electron distribution, the reduction of molecular symmetry, and the change of the charge transfer direction, **2** emitted strong NIR fluorescence. With the advantages of a good imaging effect, high photostability, and low cytotoxicity, BODIPY **2** could be applied for biological imaging. Fluorescence images of stained HeLa cells and human dental pulp cells showed that BODIPY **2** generally aggregated in the cytoplasm (red region), not in the nucleus (black region) (Figure 3).

A year later, Jiang and co-workers reported two novel fluorescent BODIPY dyes, **3** and **4**, for mitochondrial imaging (Figure 3a) [21]. Lipophilic BODIPY **3** was afforded through a Knoevenagel condensation of 4-dimethylaminobenzaldehyde with the corresponding BODIPY. Water-soluble BODIPY **4** was afforded through methylation of **3** with methyl iodide (Figure 3b). MTT experiments showed that the **3** and **4** have good biocompatibility. Confocal images indicated that **4** could target the mitochondrial region in HeLa cells specifically by introducing the cationic moiety, the TPP^+^ moiety, whereas weak fluorescence of **3** in HeLa cells was observed, which suggested that **3** was not suitable for targeting the mitochondrial region, due to its aggregation in the aqueous media (Figure 4). Both dyes inherited the good optical properties of the BODIPY core, including high photostability and fluorescence quantum yield.

### 2.2. Other Nitrogen-Containing Boron Compounds

The *N, N*-chelate boron compounds contain BOIMPY, (aza)BODIPY, BOPHY, diazaborepin, BOPYIN, etc., according to the different numbers of membered rings [22]. Meanwhile, there are some other nitrogen-containing boron compounds that could be applied for cell imaging besides BODIPY.

In 2019, Zhang et al. reported various BOPYIN derivatives, **5a**–**e**, that have been synthesized via a one-pot method [22]. Various 3,3-dimethyl-2-[2-(2-pyr-rolyl)ethenyl] indoles were formed by different 2,3,3-trimethylindole hydrochlorides with pyrrole-2-carboxaldehyde. The *N, N*-ligand compounds then reacted with BF_3_·OEt_2_ to form the BOPYIN derivatives **5a**–**e** (Figure 4). These BOPYIN derivatives were applied as biocompatible fluorophores in cell bioimaging. HeLa cells were stained with BOPYINsxxx **5a**–**e**, and these dyes rapidly passed through cell membranes, located in the cytoplasm predominantly afterwards, and showed bright green fluorescence. Confocal fluorescence images suggested that these BOPYIN were located in the perinuclear region, probably due to endocytic vesicles (Figure 5).

In 2020, Curiel et al. reported a novel four-coordinate *N, N*-difluoroboryl complex **6** (Figure 5) [23]. The compound **6** was synthesized through a substitution reaction, a Buchwald−Hartwig reaction, an intramolecular cyclization, and a B-N bond coupling, respectively. This *N, N*-difluoroboryl complex **6** has a large Stokes shift due to its unique structural features, including desymmetrization, rigidification of the ligand, and π-expansion of the conjugated system, which could be applied as a fluorescent probe for cancer cells’ imaging (Figure 6).

In 2020, Chen’s group reported various novel *N, N*-difluoroboryl complexes with a tetrahydro-quinoxaline moiety donor, **7** [24]. For the synthesis of *N, N*-difluoroboryl complexes **7** (Figure 6), a condensation of 1,4-diethyl-7-hydroxy-1,2,3,4-tetrahydroquinoxaline-6-carbaldehyde with various substituted anilines was carried out to afford the intermediates. The intermediates were subsequently treated with BF_3_·Et_2_O to obtain the corresponding products. MTT assays indicated that compound **7a** was less toxic. The HeLa cells were stained with **7a** with commercial Hoechst 33342 as a nuclear stain and bright red fluorescence could be observed from the cytoplasm (Figure 7). A 3-day-old zebrafish was fed with **7a**, and remarkable red fluorescence could be observed (Figure 7). The tetrahydro-quinoxaline donor was introduced into these D–A-type fluorescent dyes, which enhanced the intramolecular change transfer (ICT) effect, thus resulting in red shifts of the emissions and large Stokes shifts. Meanwhile, a reasonable fluorescence quantum yield was retained due to the rigidity of the tetrahydro-quinoxaline moiety as the electron donor.

## 3. Cyanine

Cyanine is a conjugated system between two nitrogen atoms; in each resonance structure, exactly one nitrogen atom is oxidized to an iminium. Cyanine is one of the most significant fluorescent dyes due to its unique optical properties, including a high molar extinction coefficient, high fluorescence quantum yield, narrow absorption/emission band, and readily tunable fluorescence profile from the UV-vis to near-infrared (NIR) range [25,26].

In 2018, Jose et al. synthesized a novel dye, **8** [27], which is a combination of heptamethine cyanine dye IR-786 [28] and an FDA-approved drug, Amoxapine [29]. IR-786 iodide 4 was generated through a methylation, a cyclization, and a dehydrative condensation, respectively [30]. Fluorescent dye **8** was synthesized through an amination reaction of IR-786 iodide with Amoxapine (Figure 7). Dye **8** could selectively stain the mitochondria, whereas IR-786 iodide does not show specificity for any organelles (Figure 8a). Colocalization experiments using different commercial mitochondrial stains incubated with dye **8** further confirmed its mitochondrial selectivity (Figure 8b). The lipophilic nature of the dye **8** could assist with penetrating the cell membrane and the delocalized positive charge helped the dye **8** to target the negatively charged mitochondria.

In 2019, Bräse’s group reported three novel polyfluorinated cyanine dyes, **9a**–**c**, and discussed their photophysical properties [31]. These cyanine probes were synthesized through a cyclization, a Heck reaction, a methylation, and a dehydrative condensation, respectively (Figure 8). All these dyes could penetrate the cell membrane and selectively accumulate in the mitochondria (Figure 9), due to the lipophilicity of cyanine dyes and the positive charge of these dyes [32]. Moreover, the addition of fluorous side chains resulted in red shifts of the absorption and emission of these cyanine dyes.

In 2022, Ge’s group reported two hemi-cyanine probes, **10a** and **10b**, and a neutral probe **10c** based on a quinoxaline skeleton [33]. The three probes were generated through a Knoevenagel condensation of 1,3-dimethylquinoxalinium iodide or 3-methylquinoxalinium with different aldehydes (Figure 9). Fluorescence imaging indicated that all these dyes could target the mitochondria of HeLa cells. Probes **10a** and **10b** stained the nucleic acid in the mitochondria with excellent selectivity, while dye **10c** neutrally stained the mitochondrial region (Figure 10). This research suggested that bioactive quinoxaline derivatives could be effectively applied in the field of fluorescence images.

Additionally, in 2022, Gallavardin and co-workers replaced the classical electron-donating group indole with indazole in merocyanines to study the substitution effect [34]. The indazole and indole merocyanines **12** and **13** were generally synthesized through a key-step Knoevenagel condensation of ethyl-3,3-dimethyl-3H-indolium iodide **11** with various corresponding aldehydes (Figure 10). These merocyanines were used to treat tumoral PC12 cells. Confocal fluorescence images indicated that indazole-based merocyanines **12a**–**c** selectively stained the mitochondria. Indole derivatives **13a**,**b** did not stain the same cell compartment: **13a** did not target the mitochondria and stained in the liposome, whereas **13b** selectively stained in the mitochondria, similar to indazole compounds (Figure 11). Indazole merocyanines **12** have an electron-donating indazole ring and a charged electron-accepting indolinium moiety to target the mitochondria, while the absorption and emission spectra of indole merocyanines **13** are slightly red-shifted because indoles are more electron-rich.

## 4. Pyridine Derivatives

Pyridine is an aromatic heterocycle composed of a six-membered ring with one nitrogen. The first pyridine was reported through heating animal bones by Scottish chemist Thomas Anderson in 1849 [35]. Due to the mitochondrial-targeting ability of the pyridinium group and other biological features of pyridine derivatives, pyridine derivatives have been generally utilized as fluorescent dyes in cellular imaging [36,37,38,39]. Herein, we mainly introduce the direct fluorescence cell imaging of pyridine derivatives.

In 2018, Zhang and Wang reported three isomers of triphenylamine-based terpyridine derivatives, **14a**–**c** [40]. These terpyridine derivatives were synthesized through an aldol reaction and a subsequent cyclization (Figure 11). The terpyridine derivatives **14a**–**c** could stain similarly in cytosolic space of the HepG2 cells instead of the nuclear region. The bright field images revealed good cell morphologies, which showed the low toxicity of **14a**–**c** (Figure 12). The D–A structure was constructed to build the optically active probes. The triphenylamine unit owned the forceful electron donor and highly efficient p-electron bridge, while the terpyridine unit was applied as the electron acceptor. Moreover, the methyl group attached to the triphenylamine unit was applied to improve the electron-donating ability and the lipophilicity.

In 2020, Huang et al. developed a novel approach to obtain C-4 arylated pyridine derivatives through a C−C coupling reaction of 2,4-dichloropyridines with boronic esters catalyzed by Palladium (Figure 12a) [41]. Here, **15a** and **15b** were chosen as representative compounds to identify their good applicability in cell imaging (Figure 12b). Blue fluorescence (Figure 13A) and green fluorescence (Figure 13B) could be observed by exciting at 405 and 488 nm, respectively, while bright cyan fluorescence was shown in merged images (Figure 13D). Inspection of the photophysical properties showed that the stronger the electron-absorbing ability of C-2 or C-4 substitution, the greater the red-shifted emission of the pyridine derivatives, indicating the formation of a D−π-A system.

In 2021, Zhou’s group synthesized various water-soluble benzoxazole-pyridinium salt derivatives [42]. These probes were generally synthesized through a bromination, a Wittig reaction, a methylation, and an ion-exchanging reaction, respectively (Figure 13). Cell imaging revealed that 3-pyridinium salt **16** crossed the nuclear membrane and selectively stained the nucleus, whereas 4-pyridinium salt derivatives **17a**–**d** stained in the nuclear membrane, which clearly showed the morphology of the nucleus (Figure 14). The plausible reason for the different staining ability may be that the nature of the pyridinium salt isomer influences the particle shape and size in aggregates in the cells.

## 5. Indole Derivatives

Indole is an aromatic heterocyclic organic compound made of benzene and a fused pyrrole ring. The first indole was generated through Fischer indole synthesis, reported in 1883 [43]. Indole derivatives have been included in the synthesis of essential FDA-approved drugs [5]. To our knowledge, the scientific researchers generally focused on indole derivatives applied as pharmaceutical drugs [44,45]. However, only a few indole derivatives applied in cell imaging systems were reported. In this section, indole derivatives for direct cell imagining were investigated.

F16 is an indole-containing, mitochondria-targeted, broad-spectrum anticancer drug. In 2018, Liu et al. reported two isomers of F16 (*o*-F16 **18b** and *m*-F16 **18c**) with different fluorescence [46]. The F16s **18a**–**c** were synthesized through a Wittig-like reaction and a methylation of gramine with the corresponding pyridine carboxaldehyde, respectively (Figure 14). Fluorescent images pointed out that the two isomers could selectively accumulate in the live SGC-7901 cells (Figure 15).

In 2019, Cheng et al. synthesized a series of representative F16 derivatives, **19**, which could selectively accumulate in the mitochondria [47]. These F16 derivatives **19** with different isomers were prepared through a Knoevenagel condensation of 1,4-dimethylpyridin-1-ium iodide with the corresponding indole-3-carboxaldehydes (Figure 15). Among them, compound **19a** showed good anti-tumor activity in various cancer cell lines. Meanwhile, **19a** could also be applied as a fluorescence probe for cancer cell imaging (Figure 16). The cell experiments revealed that the mitochondrial selectivity of F16s is driven by the negative transmembrane potential, such as other DLCs.

In 2022, Beşer et al. investigated the fluorescent probe properties of a Caulerpin derivative, **20b**, as a member of the indole family [48]. For the synthesis of this bis indole compound **20b** (Figure 16), the corresponding indole **20a** was synthesized by Fisher indolization reaction of cyclooctanone with phenylhydrazine. Afterwards, the DDQ/H_2_O/THF system selectively oxidized benzylic CH_3_ groups at the 3-position of the indole **20a** to corresponding ketone. Indolization of this ketone by another Fisher indolization reaction and a subsequent methylation furnished the bis indole **20b** in a high yield. Cytotoxic activity tests indicated the low toxicity of **20b** on various cancer and healthy cell lines. Confocal fluorescence imaging showed that the probe **20b** selectively stained the cytoplasm of MCF-7 cells (Figure 17). This research suggested that this molecule can be applied as a fluorescence probe for biological cell imaging.

## 6. Quinoline Derivatives

Quinoline is an aromatic heterocycle made of benzene and a fused pyridine ring. The first quinoline was obtained through coal tar extraction in 1834 by chemist Friedlieb Ferdinand Runge [49]. Quinoline derivatives have been widely used in various fields, including pharmaceutical, biological, and industrial chemical fields [50]. In the biological field, the quinoline derivatives have been demonstrated as star molecular probes, because of their excellent biological activities [51,52]. Herein, we focus on the synthesis and applications in direct cell imaging of quinoline derivatives.

In 2019, Chen and co-workers synthesized various novel 7-aminoquinolines **21** that exhibit high selectivity towards the Golgi apparatus [53]. These 7-aminoquinoline derivatives were synthesized through a selective condensation of m-phenylenebenzene with various 1,3-diaketones (Figure 17). Confocal images revealed that the compound **21a** chosen as a representative was located in the Golgi apparatus of various cell lines (HeLa, U2OS, and 4T1 cells), specifically (Figure 18). The strong electron-withdrawing trifluoromethyl group potentially enhances the intramolecular change transfer (ICT) state of the 7-aminoquolines between the strong electron-donating amine group and the trifluoromethyl group, thus resulting in red shifts of the absorption and emission of the compounds.

In 2020, Tian et al. synthesized various water-soluble quinolineindole-based derivatives **22a**–**c** [54]. These probes were generally synthesized through a Knoevenagel condensation of 4-methylquinolinium salt derivatives with 1*H*-indole-3-carbaldehyde (Figure 18). Confocal images suggested that **22c** as a representative could target the nucleus and mitochondria in live and fixed cells (Figure 19A,C). Colocalization experiments using different commercial nuclear stains incubated with **22c** further confirmed the nuclear and mitochondrial selectivity of **22c** (Figure 19B,D–F). I^−^ or SO_3_^−^ replaced by NO_3_^−^ on **22c** enhanced the water solubility and biocompatibility compared to **22a**,**b**.

In 2022, Ge’s group synthesized various functional dyes **23a**−**d** with a chromeno[*b*]quinoline skeleton through a cyclization of coumarin derivatives with aromatic amines in the presence of the catalyst anhydrous AlCl_3_ (Figure 19) [55]. Moreover, the optical performance, toxicity, cell imaging, and calculations of these dyes were comprehensively evaluated. Among these functional dyes, probes **23a**−**c**, which possess a diethylamine group as a chromophore, exhibited ideal fluorescence performance, and could be applied as fluorescent markers to lipid droplets in HeLa cells (Figure 20). Compared with probes **23a**−**c**, the structure of probe **23d** may lack the diethylamino group to reduce its lipophilicity, thus resulting in the inability to penetrate the organelle membrane.

## 7. Maleimide Derivatives

Maleimides are generally synthesized by dehydration of maleic anhydride with amines [56]. Maleimide derivatives have exhibited various biological properties which can be applied in medicinal and natural product chemical fields [57,58]. In the bioimaging field, due to the fluorescence properties of the maleimide group, maleimide derivatives have been employed as fluorescence probes for biological cell imaging [59,60]. Here, the direct cellular fluorescence imaging of maleimide derivatives is discussed.

In 2021, Patel et al. synthesized a series of maleimide derivatives with bright fluorescence [61]. These compounds were obtained through the tandem intra- and inter-molecular cyclization of *o*-alkynylanilines with maleimides (Figure 20). Confocal images revealed bright fluorescence of HeLa cells stained with the representative compounds **24a**–**d** (Figure 21).

Recently, Amarante and co-workers reported a transition metal-free approach to obtain maleimide derivative **25** through a novel rearrangement from thiazolidine-2,4-diones in one step [62]. *N*-butylmaleimide **26** was obtained through a substitution reaction of compound **25** and 1-bromobutane (Figure 21). Compounds **25** and **26**, which emitted green and red fluorescence, were chosen as respective compounds to study. Confocal images indicated that compound **25** could stain the cytoplasm in live and fixed cells. A mild staining in the cell nucleus could be observed (Figure 22A−C). Compound **26** stained the perinuclear region preferentially, perhaps due to its accumulation in the mitochondria (Figure 22D−F).

## 8. Others

There have been some other nitrogen-containing heterocycles with bioimaging functions reported in recent years. These *N*-heterocycles containing different amounts of nitrogen have various biological features. However, in recent years, they have not been systematically studied. In this section, we summarize the direct cell imaging of these nitrogen-containing heterocycles based on the amount of nitrogen for reference.

In 2018, Yu’s group reported various novel purine-based AIEgens **27a**–**e**, which were generated via the Suzuki coupling reaction (Figure 22) [63]. The compounds have the advantages of high selectivity, low background, and good biocompatibility. Cell imaging experiments showed that these probes could selectively stain lipid droplets and have good photostability, similar to the commercial dyes (Figure 23). A D–p-A structure was constructed to build the tunable emission AIE fluorophores. Purine was chosen as the core structure, indole was chosen as the electron donor, and the electron acceptor was changed to regulate the emission. Meanwhile, an *n*-propyl group was introduced to improve the lipophilicity.

In 2019, based on Namba’s group’s work [64], Suzenet et al. reported substituted triazapentalenes **28** which exhibited good fluorescent properties [65]. These novel triazapentalene compounds were synthesized through a one-step cyclization and a Suzuki cross-coupling reaction (Figure 23). These fluorescent probes have the advantages of high quantum yields, good photostability, and large Stokes shifts, which is suitable for optical imaging applications. Photobleaching experiments suggested that representative compound **28a** shows good photostability (Figure 24). The fused diazine on the triazapentalene ring induced strong red shifts of the emission and increased quantum yields due to a plausible ICT process.

Imidazoles are ubiquitous *N*-containing heterocyclic molecules in natural products and drugs [66,67]. In 2020, Banerji and co-workers synthesized substituted imidazoles **29** through an oxidative cyclization with the diketone or α-hydroxy ketone, aromatic aldehyde, and amine source catalyzed by iodine (Figure 24) [68]. This methodology has advantages of being peroxide-, transition metal-, and organic solvent-free. It could be employed at the gram-scale level. Due to the excellent fluorescence properties of these molecules, two of the derivatized imidazoles, **29a** and **29b**, were modified with lysosome-directing groups. These two molecules showed bright blue fluorescence of lysosomes in human and murine cells, which could be applied as lysosome-targeted probes (Figure 25).

In 2020, Yagishita et al. synthesized various novel quaternized imidazo[1,2-*a*]pyridine dyes **32a**–**d**, excited with blue light [69]. For the synthesis of these compounds (Figure 25), 2-iodinated imidazo[1,2-*a*]pyridine **30** was synthesized through a copper-catalyzed oxidative coupling of 2-aminopyridine with phenyl acetylene in the presence of I_2_ [70]. Afterwards, compound **31** was generated through a Sonogashira coupling reaction of compound **30** with 1-ethynyl-4-methoxybenzene. The alkylation of **31** with different alkyl iodides yielded various corresponding products, **32a**–**d**. The HeLa cells were stained with the compound **35a**, and colocalization experiments using a commercial mitochondrial marker, further confirming that representative salt **32a** could target the mitochondria (Figure 26). The D−π-A structure plays a key role for the two- and three-photon fluorescence imaging. The *p*-methoxyphenyl ring was chosen as the electron donor and the cationic imidazo[1,2-*a*]pyridine was chosen as the electron acceptor.

In 2021, Ornelas and co-workers reported the imidazo[1,2-*a*]pyrimidine compounds which could be applied for fluorescence imaging and PDT [71]. The alkoxylated alcohols **33a** and **33b** were obtained through a condensation, a one-step cyclization, and a Williamson reaction, respectively (Figure 26). Fluorescence images indicated the low toxicity of representative compound **33a** in the dark and showed bright fluorescence in the intracellular medium of MCF-7 cells (Figure 27). Compared with the imidazopyridine **33a**, the third nitrogen atom in the imidazopyrimidine **33b** is important for the solubility in the cell culture medium as well as the cell penetration process.

Additionally, in 2021, Klymchenko et al. reported a series of fluorescent probes **34** based on Nile Red for specific targeting of different organelle, including the endoplasmic reticulum, Golgi apparatus, lysosomes, lipid droplets, mitochondria, and plasma membranes [72]. These organelle markers were generally synthesized through a one-step amidation reaction (Figure 27). These probes were incubated with live KB cells with different commercial markers and could show significant colocalization with the corresponding markers (Figure 28).

Recently, Lavis et al. reported a series of novel mitochondrial stains **35a**–**f** based on 2,7-diaminobenzopyrylium (DAB) dyes excited with violet light [73]. To obtain the corresponding probes (Figure 28), different known 2,7-diaminobenzopyrylium (DAB) dyes were treated with Et_3_OBF_4_ to generate 2-ethoxychromenylium intermediates, and the intermediates were reacted with diethylamine afterwards. These probes could stain in mitochondria because of their positively charged scaffolds, while the julolidine-based derivatives **35c**–**e** showed brighter fluorescence due to increased lipophilicity of the compact cationic structure. DAB **35e** and diDAB **35f** showed excellent mitochondrial-targeting ability (Figure 29a). After media exchange washes, the DAB **31e** signal rapidly decreased, whereas the diDAB **35f** was retained (Figure 29b).

## 9. Conclusions

The small molecules applied for direct fluorescence cell imaging are summarized in Table 1 according to different scaffolds of nitrogen-containing heterocycles for clarity.

In summary, nitrogen-containing heterocycles play an important role in chemical and biological fields, which has attracted scientific researchers to develop new approaches for their synthesis. In this review, the synthesis and application in direct fluorescence cell imaging of *N*-heterocyclic organic small molecules were described. In organic synthesis, many classical reactions to obtain *N*-heterocycles were applied, and meanwhile, various novel approaches were developed. In biological imaging, fluorescence imaging for direct staining of the live cells, cytoplasm, or various organelles was described.

Looking forward, there remain some challenges to the direct fluorescence cell imaging of nitrogen-containing heterocycles, including: (1) How to develop more efficient and green approaches to synthesize desired nitrogen-containing heterocycles in organic synthesis. (2) It is still a challenge to develop *N*-heterocyclic dyes with high photostability, high-fluorescence quantum yields, good water-solubility, and good biocompatibility through molecular structural modification. (3) The live-cell fluorescence imaging of *N*-heterocyclic organic small molecules in the NIR region is rarely developed. Regarding these challenges, we hope that the summarization of live-cell fluorescence imaging of *N*-heterocyclic organic small molecules in this review will provide useful guidance and enlightenment for further development of fluorescence cell imaging.

## Data Availability

Not applicable.

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
