# Peer review of "Recent Advances in N-Heterocyclic Small Molecules for Synthesis and Application in Direct Fluorescence Cell Imaging"

_molecules, 2023, doi:10.3390/molecules28020733_

Round 1
Reviewer 1 Report
This manuscript by Li et al. summarizes the recent advances in N-heterocyclic small molecules for synthesis and application in direct fluorescence cell imaging. I suggest a minor revision.
1. All small molecules mentioned in the manuscript should be summarized in a table for clarity.
2. More discussions regarding the structure-property relationship should be added in the manuscript.
3. Figure 1 is too simple. I suggest authors to modify it to be more attractive. For example, adding some chemical structures and representative pictures.
Author Response
- All small molecules mentioned in the manuscript should be summarized in a table for clarity.
Answer: Thank you for your nice reminder, in page 38, line 560, “The small molecules mentioned were summarized in Table 1 according to different scaffolds of nitrogen-containing heterocycles for clarity.” and Table 1 were added.
- More discussions regarding the structure-property relationship should be added in the manuscript.
Answer: More discussions of work mechanism were added in the manuscript.
In page 3, line 73, “The BODIPY core was expanded with nearly planar annulate indoles, thus resulted in a well extended π-conjugation and a red shift of the absorption and emission.” was added.
In page 3, line 83, “the expansion of electron distribution, the reduction of molecular symmetry, and the change of charge transfer direction,” was added.
In page 4, line 102, “by introducing the cationic moiety, the TPP+ moiety” and “Both the two dyes inherited the good optical properties of the BODIPY core, including high photostability and fluorescence quantum yield.” were added.
In page 6, line 136, “due to its unique structural features including desymmetrization, rigidification of the ligand, and π-expansion of the conjugated system” was added.
In page 7, line 155, “The tetrahydro-quinoxaline donor was introduced into these D–A type fluorescent dyes, which enhanced the intramolecular change transfer (ICT) effect, thus resulted in red shifts of the emissions and large Stokes shifts. Meanwhile, a reasonable fluorescence quantum yield was retained due to the rigidity of the tetrahydro-quinoxaline moiety as the electron donor.” was added.
In page 8, line 180, “The lipophilic nature of the dye 8 could assist with penetrate the cell membrane and the delocalized positive charge helped the dye 8 to target the negatively charged mitochondria.” was added.
In page 9, line 194, “Moreover, the addition of fluorous side chains resulted in red shifts of the absorption and emission of these cyanine dyes.” was added.
In page 11, line 229, “Indazole merocyanines 12 have an electron-donating indazole ring and a charged electron-accepting indolinium moiety to target mitochondria. While the absorption and emission spectra of indole merocyanines 13 are slightly red-shifted because indoles are more electron rich.” was added.
In page 13, line 254, “The D-A structure was constructed to build the optically active probes. The triphenylamine unit owned the forceful electron donor and highly-efficient p-electron bridge, while the terpyridine unit was applicated as the electron acceptor. Moreover, the methyl group attached to the triphenylamine unit was applicated to improve the electron-donating ability and the lipophilicity.” was added.
In page 14, line 268, “Inspection of the photophysical properties showed that the stronger the electron-absorbing ability of C-2 or C-4 substitution, the greater the red-shifted emission of the pyridine derivatives, indicating the formation of a D−π−A system.” was added.
In page 16, line 284, “The plausible reason of the different stain ability may be that the nature of the pyridinium salt isomer influence the particle shape and size in aggregates in the cells.” was added.
In page 16, line 318, “The cell experiments revealed that the mitochondrial selectivity of F16s is driven by the negative transmembrane potential like other DLCs.” was added.
In page 20, line 356, “The strong electron-withdrawing trifluoromethyl group potentially enhances intramolecular change transfer (ICT) state of the 7-aminoquolines between the strong electron-donating amine group and the trifluoromethyl group, thus resulted in red shifts of the absorption and emission of the compounds.” was added.
In page 21, line 373, “I- or SO3- replaced by NO3- on 22c enhanced the water solubility and biocompatibility compared to 22a-b.” was added.
In page 23, line 388, “Compared with probes 23a−c, the structure of probe 23d may lack the diethylamino group to reduce its lipophilicity, thus resulted in the inability to penetrate the organelle membrane.” was added.
In page 27, line 452, “A D–p–A structure was constructed to build the tunable emission AIE fluorophores. Purine was chosen as the core structure; indole was chosen as the electron donor and the electron acceptor was changed to regulate the emission. Meanwhile, an n-propyl group was introduced to improve the lipophilicity.” was added.
In page 30, line 470, “The fused diazine on the triazapentalene ring induced strong red shifts of the emission and increased quantum yields due to a plausible ICT process.” was added.
In page 32, line 507, “The D−π−A structure plays a key role for the two‐ and three‐photon fluorescence imaging. The p‐methoxyphenyl ring was chosen as the electron donor and the cationic imidazo[1,2‐a]pyridine was chosen as the electron acceptor.” was added.
In page 33, line 521, “Compared with the imidazopyridine 33a, the third nitrogen atom in the imidazopyrimidine 33b is important for the solubility in the cell culture medium as well as the cell penetration process.” was added.
In page 35, line 551, “because of their positively charged scaffolds” was added.
- Figure 1 is too simple. I suggest authors to modify it to be more attractive. For example, adding some chemical structures and representative pictures.
Answer: In page 1, line 55, Figure 1 was changed, some chemical structures and representative pictures were added in Figure 1.
Reviewer 2 Report
The authors described the recent advances in N-heterocyclic small molecules for synthesis and application in direct fluorescence cell imaging. I consider that review meets all requirements to be published in Molecules (Q1) in terms of high scientific rigor and relevance of the information. Some suggestions are included: (1) See Schemes. The yield of each step might be included. It is not unified. (2) See Figures. If the journal is not open access, the copyright right permission CCC might be included. (3) See Figures. The resolution should be improved. For instance, see Figure 12. (4) See Scheme 12a. The quality should be improved. (5) See line 339. The letter “H” might be in italic. It should be unified in the manuscript. (6) See Schemes 21 and 24. It should be unified like the other schemes. (7) The type of cancer for each cell line should be included in parenthesis. Also, the first letter should be capitalized. For instance, HeLa cells (Leukemia) instead of HeLa cells (See line 73). It should be modified in the manuscript.Author Response
(1) See Schemes. The yield of each step might be included. It is not unified.
Answer: In Schemes, the yield of each step was included.
(2) See Figures. If the journal is not open access, the copyright right permission CCC might be included.
Answer: The copyright permissions of Figures and Schemes were uploaded.
(3) See Figures. The resolution should be improved. For instance, see Figure 12.
Answer: Figure 12, Figure 23, Figure 25 were changed.
(4) See Scheme 12a. The quality should be improved.
Answer: Scheme 12a was changed.
(5) See line 339. The letter “H” might be in italic. It should be unified in the manuscript.
Answer: In line 339, the letter “H” was in italic.
(6) See Schemes 21 and 24. It should be unified like the other schemes.
Answer: Schemes 21 and 24 were changed.
(7) The type of cancer for each cell line should be included in parenthesis. Also, the first letter should be capitalized. For instance, HeLa cells (Leukemia) instead of HeLa cells (See line 73). It should be modified in the manuscript.
Answer: In line 73, “HeLa cells” were cited according to the reference 19, however, “HeLa cells (Leukemia)” were not mentioned in the reference 19.
Reviewer 3 Report
1. Although the authors provided a number of examples of how N-heterocyclic compounds may be used in fluorescent cell imaging, they were often unclear on why or how the molecules really worked in fluorescent cell imaging.
2. Improvements to the English language are essential. More than fifteen times throughout the essay, for instance, the term "indicated" appears. Also, it should be "contain" instead of "contains" in line 108.
3. I believe the affiliations should be rearranged to reflect the authors' order.
Author Response
- Although the authors provided a number of examples of how N-heterocyclic compounds may be used in fluorescent cell imaging, they were often unclear on why or how the molecules really worked in fluorescent cell imaging.
Answer: More discussions of work mechanism were added in the manuscript.
In page 3, line 73, “The BODIPY core was expanded with nearly planar annulate indoles, thus resulted in a well extended π-conjugation and a red shift of the absorption and emission.” was added.
In page 3, line 83, “the expansion of electron distribution, the reduction of molecular symmetry, and the change of charge transfer direction,” was added.
In page 4, line 102, “by introducing the cationic moiety, the TPP+ moiety” and “Both the two dyes inherited the good optical properties of the BODIPY core, including high photostability and fluorescence quantum yield.” were added.
In page 6, line 136, “due to its unique structural features including desymmetrization, rigidification of the ligand, and π-expansion of the conjugated system” was added.
In page 7, line 155, “The tetrahydro-quinoxaline donor was introduced into these D–A type fluorescent dyes, which enhanced the intramolecular change transfer (ICT) effect, thus resulted in red shifts of the emissions and large Stokes shifts. Meanwhile, a reasonable fluorescence quantum yield was retained due to the rigidity of the tetrahydro-quinoxaline moiety as the electron donor.” was added.
In page 8, line 180, “The lipophilic nature of the dye 8 could assist with penetrate the cell membrane and the delocalized positive charge helped the dye 8 to target the negatively charged mitochondria.” was added.
In page 9, line 194, “Moreover, the addition of fluorous side chains resulted in red shifts of the absorption and emission of these cyanine dyes.” was added.
In page 11, line 229, “Indazole merocyanines 12 have an electron-donating indazole ring and a charged electron-accepting indolinium moiety to target mitochondria. While the absorption and emission spectra of indole merocyanines 13 are slightly red-shifted because indoles are more electron rich.” was added.
In page 13, line 254, “The D-A structure was constructed to build the optically active probes. The triphenylamine unit owned the forceful electron donor and highly-efficient p-electron bridge, while the terpyridine unit was applicated as the electron acceptor. Moreover, the methyl group attached to the triphenylamine unit was applicated to improve the electron-donating ability and the lipophilicity.” was added.
In page 14, line 268, “Inspection of the photophysical properties showed that the stronger the electron-absorbing ability of C-2 or C-4 substitution, the greater the red-shifted emission of the pyridine derivatives, indicating the formation of a D−π−A system.” was added.
In page 16, line 284, “The plausible reason of the different stain ability may be that the nature of the pyridinium salt isomer influence the particle shape and size in aggregates in the cells.” was added.
In page 16, line 318, “The cell experiments revealed that the mitochondrial selectivity of F16s is driven by the negative transmembrane potential like other DLCs.” was added.
In page 20, line 356, “The strong electron-withdrawing trifluoromethyl group potentially enhances intramolecular change transfer (ICT) state of the 7-aminoquolines between the strong electron-donating amine group and the trifluoromethyl group, thus resulted in red shifts of the absorption and emission of the compounds.” was added.
In page 21, line 373, “I- or SO3- replaced by NO3- on 22c enhanced the water solubility and biocompatibility compared to 22a-b.” was added.
In page 23, line 388, “Compared with probes 23a−c, the structure of probe 23d may lack the diethylamino group to reduce its lipophilicity, thus resulted in the inability to penetrate the organelle membrane.” was added.
In page 27, line 452, “A D–p–A structure was constructed to build the tunable emission AIE fluorophores. Purine was chosen as the core structure; indole was chosen as the electron donor and the electron acceptor was changed to regulate the emission. Meanwhile, an n-propyl group was introduced to improve the lipophilicity.” was added.
In page 30, line 470, “The fused diazine on the triazapentalene ring induced strong red shifts of the emission and increased quantum yields due to a plausible ICT process.” was added.
In page 32, line 507, “The D−π−A structure plays a key role for the two‐ and three‐photon fluorescence imaging. The p‐methoxyphenyl ring was chosen as the electron donor and the cationic imidazo[1,2‐a]pyridine was chosen as the electron acceptor.” was added.
In page 33, line 521, “Compared with the imidazopyridine 33a, the third nitrogen atom in the imidazopyrimidine 33b is important for the solubility in the cell culture medium as well as the cell penetration process.” was added.
In page 35, line 551, “because of their positively charged scaffolds” was added.
- Improvements to the English language are essential. More than fifteen times throughout the essay, for instance, the term "indicated" appears. Also, it should be "contain" instead of "contains" in line 108.
Answer:
English language was checked.
In page 3, line 87, “indicated” was changed to “showed”.
In page 4, line 103, “indicated” was changed to “suggested”.
In page 5, line 126, “indicated” was changed to “suggested”.
In page 9, line 192, “Fluorescence images indicated that all” was changed to “All”.
In page 10, line 209, “indicated” was changed to “suggested”.
In page 10, line 209, “The fluorescence images stained with indole derivatives 13a and 13b revealed that they” was changed to “Indole derivatives 13a-b”
In page 12, line 252, “indicated” was changed to “showed”.
In page 17, line 303, “indicated” was changed to “pointed out that”.
In page 19, line 334, “indicated” was changed to “showed”.
In page 19, line 335, “indicated” was changed to “suggested”.
In page 21, line 369, “indicated” was changed to “suggested”.
In page 27, line 450, “indicated” was changed to “showed”.
In page 30, line 469, “indicated” was changed to “suggested”.
In page 5, line 114, “contains” was changed to “contain”
- I believe the affiliations should be rearranged to reflect the authors' order.
Answer: The authors of the cited references were different generally. The affiliations were arranged according to the scaffolds and years.
Round 2
Reviewer 3 Report
The manuscript can be accepted for publication in the current form.
Author Response
Thank you for your advice.
